# PePR: Performance Per Resource Unit as a Metric to Promote Small-Scale Deep Learning in Medical Image Analysis

Raghavendra Selvan[*1], Bob Pepin[1], Christian Igel[1], Gabrielle Samuel[2], and Erik B Dam[1]

[1]Dept. of Computer Science , University of Copenhagen, Copenhagen, Denmark
[2]Dept. of Global Health , King's College London , London, United Kingdom
{raghav, bope,igel,erikdam}@di.ku.dk, gabrielle.samuel@kcl.ac.uk

## Abstract

The recent advances in deep learning (DL) have been accelerated by access to large-scale data and compute. These large-scale resources have been used to train progressively larger models which are resource intensive in terms of compute, data, energy, and carbon emissions. These costs are becoming a new type of entry barrier to researchers and practitioners with limited access to resources at such scale, particularly in the *Global South*. In this work, we take a comprehensive look at the landscape of existing DL models for medical image analysis tasks and demonstrate their usefulness in settings where resources are limited. To account for the resource consumption of DL models, we introduce a novel measure to estimate the performance per resource unit, which we call the PePR[1] score. Using a diverse family of 131 unique DL architectures (spanning $1M$ to $130M$ trainable parameters) and three medical image datasets, we capture trends about the performance-resource trade-offs. In applications like medical image analysis, we argue that small-scale, specialized models are better than striving for large-scale models. Furthermore, we show that using existing pretrained models that are fine-tuned on new data can significantly reduce the computational resources and data required compared to training models from scratch. We hope this work will encourage the community to focus on improving AI equity by developing methods and models with smaller resource footprints.[2]

## 1 Introduction

The question of material costs of technology, even in light of their usefulness, should not be ignored [1]. This is also true for technologies such as deep learning (DL) that is reliant on large-scale data and compute, resulting in increasing energy consumption and corresponding carbon emissions [2]. These growing resource costs can hamper their environmental and social sustainability. [3, 4].

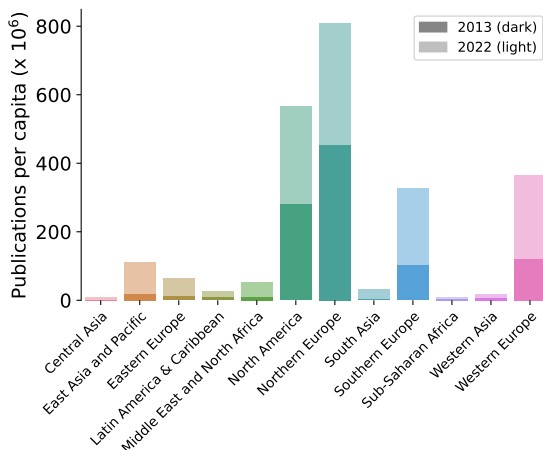

**Figure 1.** Number of publications per capita in different regions of the world for 2013 and 2022 on the topic broadly seen as "Artificial Intelligence". A large gap continues to persist in regions from the *Global South* compared to other well-performing regions, primarily in the *Global North* . Data source: OECD.ai

Considerations towards improving the environmental impact of DL are garnering attention across different application domains. This has resulted in calls for action broadly, and also within the medical image analysis community, to improve the resource efficiency of DL models [5, 6] and to report the energy and carbon costs [7]. Another important implication of the growing resource costs of DL is the risk of disenfranchising practitioners with limited access to resources. This is captured in Figure 1 which shows the number of publications (per capita) within DL across the world for 2013 and 2022[3]. Several regions in the world categorised as *Global South* are continuing to lag behind in research in DL [8]. While there are also multiple other structural reasons for this trend, the increasing resource costs of perform-

---

[*]Corresponding Author.
[1]Pronounced *pepper*.
[2]Source code: https://github.com/saintslab/PePR.

[3]The data for the visualisation in Figure 1 was curated by querying for the number of research publications per country on the topic of "Artificial Intelligence" in OpenAlex.org. The population data per country was queried from data.WorldBank.org. Regional aggregation was performed using OECD standards and further refined into the ten regions. Curated data will be provided along with the source code.

ing research within DL can become a new form of entry barrier that can aggravate this disparity [9].

In light of these observations, this work argues for focusing on small-scale DL in the era of large-scale DL. We hypothesize that the current situation with the increasing resource consumption is due to the singular focus on task-specific performance metrics that are not grounded in material costs. We also argue that access is a prerequisite to improving equity in DL and in use of these methods in healthcare. These arguments are supported by a comprehensive analysis of performance and resource costs of DL-based computer vision models. We study the properties of 131 models ranging from $1M$ to $130M$ trainable parameters, on three medical image classification tasks to capture interesting trends. We provide qualitative evidence for the usefulness of using pretrained models in resource-constrained regimes. Finally, we present a novel composite measure of performance and resource consumption. We call this the performance per resource unit (PePR) score. Using the PePR-score we characterise the behaviour of small-scale and large-scale DL models. We demonstrate that in resource-constrained regimes, small-scale DL models yield a better trade-off between performance and resource consumption.

**Related Work:** Pareto optimisation of performance and resource constraints has been primarily investigated within the context of neural architecture search (NAS) [10]. More recently, methods have been proposed to explore models using specific resource constraints such as energy consumption [11, 12] or carbon footprint [13]. The work in [11] proposes a resource-aware performance metric similar to our contribution in this work which, however, is concerned with non DL models. Within application domains such as medical image analysis, there has been little emphasis on the joint optimisation of performance and energy consumption [14]. The question of equitable AI within healthcare has been posed in works like [15] primarily from the context of fairness and not from resource/access perspectives.

## 2  PePR-score

In this work, we assume a DL model to be an entity that consumes resources such as data, energy, time, or CO2eq. budget as input and provides some measurable predictive performance on downstream tasks of interest. In contrast to conventional performance metrics that are not grounded in material costs, we envision a score that can take the resource costs into account. To this end, we introduce the notion of performance per resource unit (PePR), denoted as $P_{\mathrm{ePR}} : [0,1] \times [0,1] \to \mathbb{R}$, which relates (normalised) performance $P \in [0,1]$ of a model with the resources consumed and defined as

$$P_{\mathrm{ePR}}\left(R, P\right) = \frac{P}{1 + R}. \qquad (1)$$

In this definition, $R$ is the resource cost normalised to lie in $[0,1]$, or explicitly $R = (R_{\mathrm{abs}} - R_{\mathrm{min}})/(R_{\mathrm{max}} - R_{\mathrm{min}})$ for some absolute resource cost $R_{\mathrm{abs}}$ and some $R_{\mathrm{min}}, R_{\mathrm{max}}$ fixed across models within an experiment.[4]

The salient features of the PePR-score that make it useful as an alternative objective that takes resource costs into account are as follows:

1. **Performance-dependent sensitivity:** From the plot of the PePR isoclines (see Figure 2-a)), it is clear that PePR is insensitive to resource consumption for models with low performance. For models with high performance, PePR attributes almost identical weight to performance and to resource consumption.

2. **PePR-score for a single model:** PePR score is a relative measure of performance-resource consumption trade-off. In instances where a single model is considered, it is the same as performance. This is due to the fact that $R_{\mathrm{min}} = R_{\mathrm{max}} \implies R = 0$ and $P_{\mathrm{ePR}}(0, P) = P$.

3. **Comparing two models:** Consider the case where only two models are compared with respective absolute resource consumptions $R_{\mathrm{abs},0}, R_{\mathrm{abs},1}$ and test performances $P_0, P_1$. If $R_{\mathrm{abs},0} < R_{\mathrm{abs},1}$, then the normalized resource costs are $R_0 = 0, R_1 = 1$ because $R_{\mathrm{min}} = R_0, R_{\mathrm{max}} = R_1$. Thus, $P_{\mathrm{ePR}}(R_0, P_0) = P_0$ and $P_{\mathrm{ePR}}(R_1, P_1) = P_1/2$.

4. **PePR-score of random guessing:** Consider a binary classification task with no class imbalance. In this setting, the performance of random guessing should be about $P = 0.5$. As the $R = 0$ for this "model", the PePR-score is the same as performance.

Depending on what resource costs are used, different variations of the PePR-score can be derived. For instance, if energy consumption is used as the cost, then $R = E$ resulting in the PePR-E score. Similarly, one can derive the PePR-C (CO2eq.), PePR-D (data), PePR-M (memory), or PePR-T (Time) scores. Idealised PePR-E scores are plotted in Figure 2-a) which captures the trade-off between performance and resource consumption. Models with low resource consumption and high performance would gravitate towards the upper left corner where the PePR score approaches unity.

We also note that in cases where performance is deemed to be more important than resource consumption, PePR score can be adjusted to reflect

---

[4]Standard scaling might not always be appropriate. Outliers may have to be considered, and in other instances $R_{\mathrm{min}}, R_{\mathrm{max}}$ might depend on the experimental set-up.

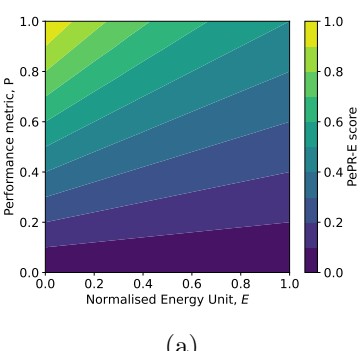
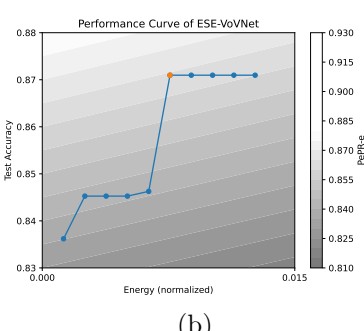
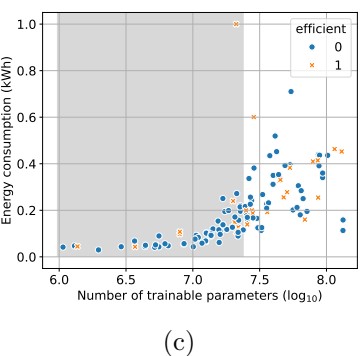

(a)  (b)  (c)

**Figure 2.** (a) Idealized PePR-E profile. (b) Performance curve for ESE-VoVNet*[16]. The orange point marks $P^*_{\text{ePRc}}$, beyond which the performance curve enters the region of diminishing returns. (c) Number of trainable parameters and energy consumption for the 131 models, demonstrating a large variability in model scale. The vertical red line demarcates the median point for number of trainable parameters.

this. For instance, one can employ $P_{\text{ePR}}(R, P; \alpha) = \alpha \cdot P/(\alpha + R)$ with a scaling factor $\alpha \geq 1$. Setting a large $\alpha$ value, say $\alpha = 100$, would prioritise performance and disregard the effect of the resource consumption. As an example, consider the PePR score for the most resource intensive model that also achieves the best performance (i.e., $P = 1.0, R = 1.0$). According to the definition in Eq. (1), the PePR score is $P_{\text{ePR}}(R = 1, P = 1; \alpha = 1) = 0.5$. Increasing the emphasis on performance using $\alpha = 100$ gives $P_{\text{ePR}}(R = 1, P = 1; \alpha = 100) = 0.99$, basically ignoring the resource costs, if the application warrants this. Adjusting $\alpha$ offers a spectrum of trade-offs between performance and resource costs. In this work, we are focussed on operating in resource constrained regimes, and are mainly interested in the setting $\alpha = 1$.

**Performance curve:** For a function $f$ representing a performance curve mapping resource costs to performance (e.g., if the resource is update steps or training data set size, it represents a rescaled learning curve), we define a PePR curve:

$$P_{\text{ePRc}}(R; f) = P_{\text{ePR}}(R, f(R)), \qquad (2)$$

where in cases of ties the smallest value is picked. Furthermore, in order to be able to compare models based on their performance curves, we define a scalar quantity $P^*_{\text{ePRc}}(f)$ by

$$P^*_{\text{ePRc}}(f) = \max_R P_{\text{ePRc}}(R; f).$$

To get some intuition on the PePR score, we can rewrite (2) as the integral of its derivative to obtain the integral representation

$$P_{\text{ePRc}}(R; f) = f(0) + \int_0^R \frac{f'(r)}{1+r} dr - \int_0^R \frac{f(r)}{(1+r)^2} dr.$$

Here, $f'$ is the derivative of $f$ with respect to resource consumption, which can be interpreted as how much of a performance increase the model is able to

get per resource consumed. First, note the presence of the weighting factors $1/(1 + r)$ and $1/(1 + r)^2$, which express that the score puts a higher weight on the performance of the model in low-resource regimes (small $r$).

Second, we can see that the score emphasizes performance per resource consumed (first integral with $f'$) and de-emphasizes absolute performance (second integral with $f$). Since all integrals are positive, the PePR score is always greater or equal to the performance of the model at zero resource consumption.

Since $f(0) \leq f(r), r \leq 1$, if we assume $f$ to be increasing, we also have that PePR increases in intervals where $f' > 1$ and decreases in intervals where $f' < f(0)/2$.[5] This captures the idea that the maxima of the PePR curve lie at points of diminishing returns as captured by $f'$, which is also visualized in Figure 2-b).

## 3 Data & Experiments

To demonstrate the usefulness of the PePR-score, we curated a large collection of diverse, neural network architectures and experiment on multiple datasets.

**Space of models:** The model space used in this work consists of 131 neural network architectures specialised for image classification. The exact number of 131 architectures was obtained after seeking sufficiently diverse models which were also pretrained on the same benchmark dataset.

We used the Pytorch Image Models (timm) model zoo to access models pretrained on ImageNet-1k resulting in 131 models spanning 1M to 131M trainable parameters. We randomly sub-sampled the available models in `Pytorch Image Models` library [17], which during our experiments had about 700 models.

---

[5]Because of the bound for the second integrand in (2): $\frac{f(0)}{2} \leq \frac{f(r)}{1+r} \leq 1$

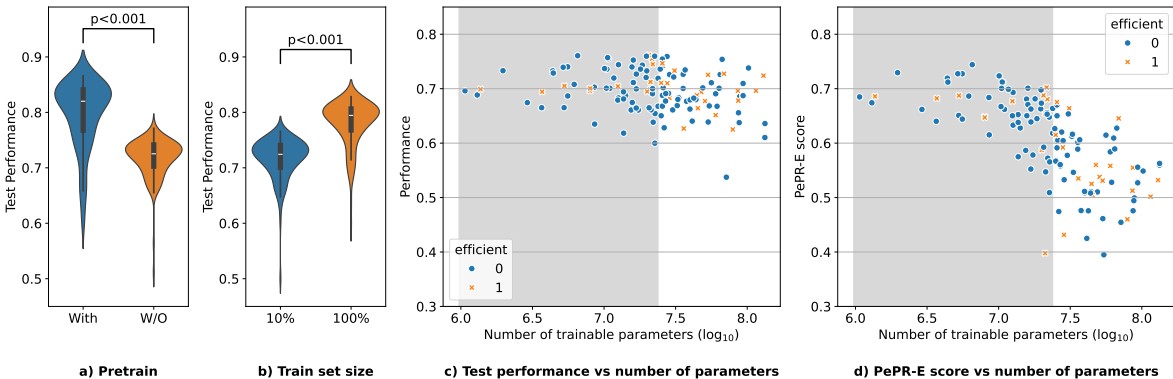

**Figure 3.** a) Violin plot showing the influence of fine-tuning the pretrained models for ten epochs versus training the models from scratch for ten epochs for all 131 models. b) Violin plot showing the influence on test performance of fine-tuning all models on 100% and 10% of training data, across all three datasets. (c) Test performance $P \in [0, 1]$ averaged over three datasets for each of the 131 models, fine-tuned for 10 epochs, against the number of trainable parameters on $\log_{10}$ scale. (d) PePR-E score for the 131 models averaged over the three datasets.

We chose as many unique architectures as possible that were all pre-trained on the same ImageNet dataset. This resulted in the 131 models used in our work, covering CNNs, vision transformers, hybrid models, and efficient architectures.

We categorise these models along two dimensions i) `CNN` or `Other` depending on if the architecture is a generic CNN primarily consisting of convolutional layers, residual connections, and other standard operators. This implies transformer-based models [18], for instance, are marked `Other` ii) `Efficient` or `Not Efficient` if the descriptions in the corresponding publications discuss any key contributions for improving some aspect of efficiency. Given these categorisations, we end up with a split of 80 and 51 for `CNN, Other`, respectively, and 31 and 100 for `Efficient, Not Efficient`, respectively. The median number of parameters is 24.6M. We further classify the models in the lower half to be *small-scale* and the upper half into *large-scale* for simplicity. The model space is illustrated in Figure 2-c) and additional details are provided for each model in Table A.1.

**Datasets:** Experiments in this work are performed on three medical image classification datasets: Derma, LIDC, Pneumonia. Derma and Pneumonia datasets are derived from the MedMNIST+ benchmark [22] and LIDC is derived from the LIDC-IDRI dataset [28]. Images in all three datasets are of $256 \times 256$ pixel resolution with intensities rescaled to $[0, 1]$. All three datasets are split into train/valid/test splits: Derma (7,007/1,003/2,005), LIDC (9,057/3,019/3,020), and Pneumonia (4,708/524/624). Derma consists of seven target classes whereas the other two datasets contain binary labels.

**Experimental design:** All models were implemented in Pytorch, trained or fine-tuned for 10 epochs with a learning rate of $5 \times 10^{-4}$ using a batch size of 32 on an Nvidia RTX3090 GPU workstation with 24 GB memory. Statistical significance is measured by $t$-tests. We considered training or fine-tuning of 10 epochs to reduce the compute resources used in this work. We expand on this choice in Sec. 4. The training of models in this work was estimated to use 58.2 kWh of electricity contributing to 3.7 kg of CO2eq. This is equivalent to about 36 km travelled by car as measured by Carbontracker [35].

**Experiments and Results:** We performed three main experiments with our large collection of models: i) Study the influence of pretraining on test performance ii) Evaluate the role of number of training points iii) Compute PePR-E score and compare the trade-off between test performance and energy consumption as the cost. Results from all three experiments are summarized in Figure 3.

We had access to pretrained weights for all 131 models, which made it possible to investigate the influence of using pretraining when resources are constrained. We either fine-tune or train-from-scratch all models for 10 epochs. In Figure 3-a), across the board, we notice that using pretrained models are significantly better compared to training models from scratch for the same number of epochs ($p < 0.001$).

Another resource that can be lacking, on top of compute/energy, is the amount of training data. We study this by only using 10% of the training data, for each of the three datasets, and reporting the average test performance per model in Figure 3-b). Even though there is a significant test performance difference ($p < 0.001$) when only using 10% of the data compared to using 100% of the data, it could be still useful in making some preliminary choices.

The overall test performance averaged across the three datasets is plotted against the number of pa-

**Table 1.** Results across all the experiments comparing the resources such as GPU memory usage in gigabyte: $\mathbf{M}_{(GB)}$, energy consumption during 10 epochs of training in watt-hour: $\mathbf{E}_{(Wh)}$, training time for 10 epochs in second: $\mathbf{T}_{(s)}$, test performance and the PePR-E score. In addition, the number of trainable parameters are also reported in million: $|\mathbf{W}|_{(M)}$. For the Derma dataset, results with no pretraining are also reported: $\mathbf{Derma}_{NPT}$. Architectures that appear more than once across the four experiments are highlighted with [*].

| Dataset | Model | Efficient | $|\mathbf{W}|_{(M)}$ | $\mathbf{M}_{(GB)}$ ↓ | $\mathbf{E}_{(Wh)}$ ↓ | $\mathbf{T}_{(s)}$ ↓ | Test P↑ | PePR-E↑ |
|---|---|---|---|---|---|---|---|---|
| $\mathbf{Derma}_{NPT}$ | ESE-VoVNet[*][16] | ✓ | 6.5 | 3.8 | 20.4 | 12.9 | 0.7651 | **0.7070** |
| | ResNet-18[*][19] | ✗ | 11.7 | 1.7 | **17.0** | **10.6** | 0.7480 | 0.7014 |
| | ResNet-34[*][19] | ✗ | 21.8 | 2.3 | 23.5 | 14.9 | 0.7617 | 0.6973 |
| | CrossVIT [20] | ✓ | 8.6 | 2.3 | 23.2 | 14.5 | 0.7550 | 0.6921 |
| | ConvNext [21] | ✗ | 3.7 | **1.6** | 18.5 | 11.9 | 0.7273 | 0.6781 |
| | HaloNet-50 [18] | ✗ | 22.7 | 7.3 | 47.7 | 29.9 | **0.7712** | 0.6498 |
| $\mathbf{Derma}$[22] | Ghostnet [23] | ✓ | 5.2 | **2.0** | 17.4 | 11.4 | 0.8579 | **0.8026** |
| | ESE-VoVNet[*][16] | ✓ | 6.5 | 3.8 | 20.4 | 12.9 | 0.8634 | 0.7992 |
| | FBNet [24] | ✓ | 5.6 | 3.1 | 18.5 | 11.9 | 0.8528 | 0.7950 |
| | MobileNetV2[*][25] | ✓ | 2.0 | 2.2 | **10.7** | **7.3** | 0.8251 | 0.7916 |
| | MNASNet100[26] | ✓ | 4.4 | 2.3 | 15.2 | 10.0 | 0.8362 | 0.7889 |
| | EdgeNext [27] | ✓ | 18.5 | 4.8 | 50.6 | 32.6 | **0.8659** | 0.7221 |
| $\mathbf{LIDC}$ [28] | MNASNet100[26] | ✓ | 4.4 | 2.4 | **18.6** | **11.7** | 0.6732 | **0.6376** |
| | ResNet-18[*][19] | ✗ | 11.7 | **1.7** | 20.4 | 12.5 | 0.6689 | 0.6303 |
| | ResNet-14 [19] | ✗ | 10.1 | 2.5 | 22.3 | 13.7 | 0.6709 | 0.6289 |
| | ResNet-34[*][19] | ✗ | 21.8 | 2.3 | 21.7 | 20.2 | 0.6868 | 0.6273 |
| | ResNet-26 [19] | ✗ | 16.0 | 3.4 | 31.0 | 19.5 | 0.6818 | 0.6240 |
| | DPN-107 [29] | ✗ | 86.9 | 16.3 | 228.0 | 138.9 | **0.6955** | 0.4133 |
| $\mathbf{Pneum.}$[22] | DLA-460 [30] | ✗ | 1.3 | 2.5 | 8.8 | 5.8 | 0.9539 | **0.9053** |
| | HardcoreNAS[31] | ✓ | 5.3 | 2.3 | 8.5 | 5.6 | 0.9523 | 0.9050 |
| | MobileNetV2[*][25] | ✓ | 2.0 | **2.2** | **5.6** | **4.0** | 0.9178 | 0.8874 |
| | MobileVitV2 [32] | ✓ | 1.4 | 3.1 | 8.6 | 5.6 | 0.9293 | 0.8828 |
| | SEMNASNet [33] | ✓ | 2.9 | 2.8 | 8.4 | 5.5 | 0.9276 | 0.8821 |
| | PNASNet [34] | ✗ | 86.1 | 22.7 | 105.9 | 64.8 | **0.9605** | 0.5830 |

rameters, along with architecture classes, in Figure 3-c). There was no significant group difference in test performance between small- and large-scale models. Similarly, there was no significant difference between models that are `Efficient` and `Not Efficient`, or between `CNN` and `Other`.

Finally, in Figure 3-d) we visualise the PePR-E score for all the models, which uses the energy consumption for fine-tuning for 10 epochs as the resource, which is then normalised within each experiment (dataset). The first striking observation is that the PePR-E scores for the larger models reduce, whereas for the smaller models there is no difference relative to other small-scale models. This is expected behaviour as PePR score is performance per resource unit, and models that consume more resources relative to other models will get a lower PePR score. We observed a significant difference in median PePR-E scores between small and large models for all three datasets, with the group of small models having a higher median PePR-E score ($p < 0.001$), shown in Figure A.1. We did not consistently observe any other significant difference across datasets in test performance or PePR-E score when stratifying by model type (`CNN` vs. `Other`) or between `Efficient` and `Not Efficient` models. Results for the top five models sorted based on their PePR-E score for each dataset along with their test performance, number of parameters, memory consumption, absolute energy consumption, training time for 10 epochs, are reported in Table 1. We also report the best per-forming model when only test performance is used as the criterion for comparison.

## 4 Discussion & Conclusion

Our experiments reported in Figure 3 and Table 1 reveal interesting trends about the interplay between test performance and resource consumption. We consider all models below the median number of parameters (24.6M) to be small-scale, and above as large-scale models, visualised demarcated using the gray-shaded regions in all relevant figures. We noticed no significant difference in performance between the small-scale and large-scale models in the regime where they were fine-tuned with pretrained weights for 10 epochs. This captures the problem with focusing only on test performance, as it could easily yield large-scale models even when small-scale models could be adequate. However, when using the PePR-E score, we see a significant performance difference with the small-scale models achieving a higher PePR-score ($p < 0.05$). This emphasises the usefulness of taking resource costs into account, which can be easily done using any variations of the PePR score.

Energy, or other resource, consumption awareness can also be incorporated using multi-objective optimisation [12]. PePR score can be thought of as one way to access the solutions on the Pareto front with an emphasis on low-resource footprint. This is cap-

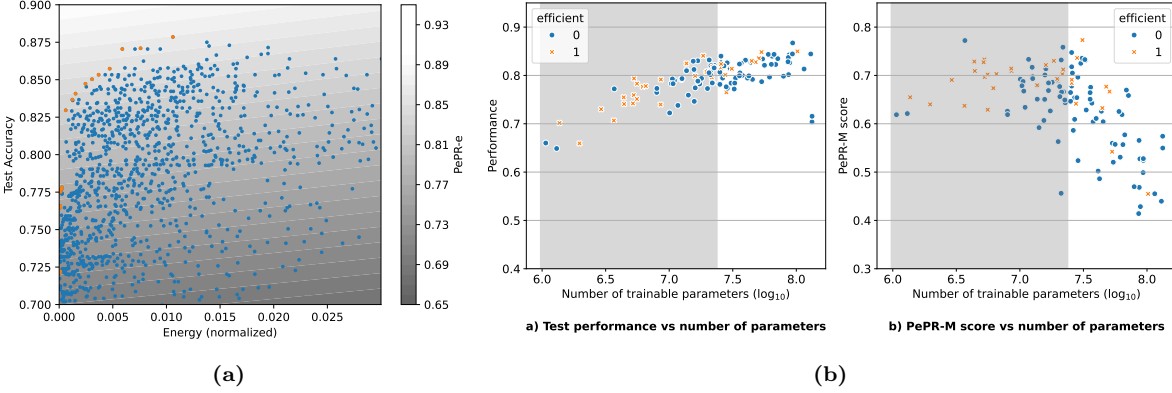

**(a)**                                                                                     **(b)**

**Figure 4.** a) Test accuracy against normalized energy used for training on Derma dataset. Points correspond to combinations of model and training epoch. Orange points lie on the Pareto frontier. Background shaded according to PePR-e score. b) Validation performance and the corresponding PePR-M scores for all models trained until convergence on ImageNet dataset using the publicly available data from [17]. PePR score shows that smaller models achieve a better performance and resource trade-off.

tured in Figure 4 which overlays the Pareto set (in orange) and all other models over the PePR scores. The knee point of this Pareto front is pointing towards maximising PePR-E score (brighter regions).

PePR score is a composite metric that offers a trade-off between performance and resource consumption. It can be used instead of multi-objective optimisation of the two objectives separately. As shown in our experiments, PePR score can be used to compare models that use different extents of resources. Current reporting in deep learning for image analysis focus on performance metrics like accuracy while disregarding the resources expended [7]. Furthermore, PePR can be used to choose the best model under a known resource constraint, such as maximum memory or energy consumption allowed.

The key experiments reported consider energy consumption as the main resource in the PePR-E score. Additional metrics (PePR-M for memory, PePR-C for carbon emissions, PePR-T for training time) reported in the Figure A.2 show the versatility of the PePR score. We can envision a general PePR score which can consider all resources into account by weighting them differently. For example, using $P_{\text{ePR}} = \frac{P}{1+\sum_i w_i R_i}$ with $\sum_i w_i = 1$, where the different weights can be adjusted depending on the application.

**Limitations:** We used a training or fine-tuning budget of 10 epochs in this work to reduce the compute resources used. This can be a limitation, as different models learn at different rates. To show that our experimental results are not artifacts of this choice, we looked at the performance of models that have been trained to convergence on ImageNet (which formed the basis of pre-training) using the public dataset from [17]. We performed a similar analysis of validation set performance of the converged models, The PePR-M scores are shown in Figure 4-b), and they show similar trends as our

experiments in Figures 3 and A.2.

The PePR score itself is agnostic to the downstream task. In this study, the experiments focussed on medical image classification, which may limit the generalisability of the results. While the findings were consistent across the considered data sets, expanding the study to other tasks (segmentation) and domains (non-image) in future work might provide further insights.

**Conclusions:** Using a large collection of DL models we have shown that using pre-trained models yields significant gains in performance, and should always be considered. We have also shown that when resource consumption is taken into account, small-scale DL models offer a better trade-off than large-scale models. Specifically, the performance achieved per unit of resource consumption for small-scale models in low-resource regimes is higher. We proposed the PePR score that offers an inbuilt trade-off between resource consumption and performance. The score penalises models with diminishing returns for a given increase in resource consumption.

Questions around how best to improve equity in research and healthcare are neither easy nor straightforward, go far beyond the ways in which we use specific types of DL, and cannot be fixed through technological solutionism [36]. Nevertheless, using small-scale DL can help mitigate certain types of inequities by reducing some of the barriers that are currently in place for researchers and practitioners with limited access to resources. Small-scale DL can be developed and run on end-point consumer hardware which is more pervasive than specialised datacenters with high performance computing in many parts of the world. With this work, we sincerely hope that by focusing on reducing the resource costs of DL to improve access the larger question of equity in DL for healthcare will be grappled with by the community.

**Acknowledgments:**
RS, BP, CI, and ED acknowledge funding received under European Union's Horizon Europe Research and Innovation programme under grant agreements No. 101070284 and No. 101070408. CI acknowledges support by the Pioneer Centre for AI, DNRF grant number P1. GS would like to acknowledge Wellcome Foundation (grant number 222180/Z/20/Z).

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

# A    Additional Results

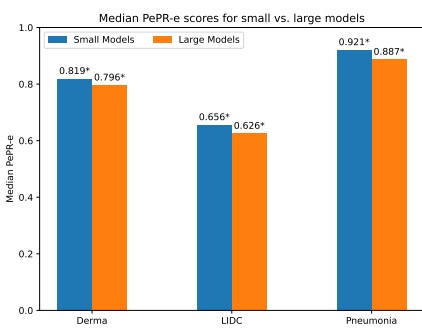

**Figure A.1.** Median PePR-e score for small models (≤ 24.6M parameters) and large models (> 24.6M parameters). All differences are significant ($p < 0.05$).

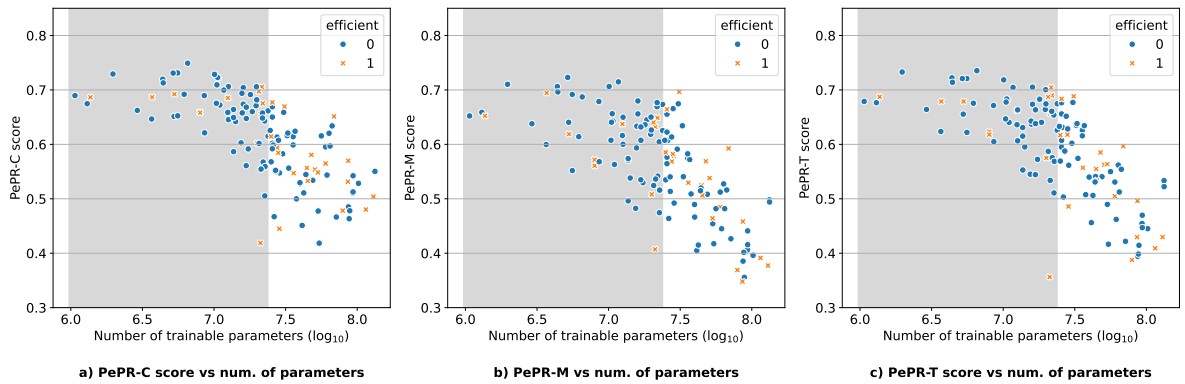

**Figure A.2.** PePR-C, PePR-M, and PePR-T scores that account for carbon emissions, GPU memory consumption and training time, respectively.

**Table A.1.** Model space used in this work described using their instance name in TIMM, number of trainable parameters, and their classifications. Models can be accessed from https://huggingface.co/models?library=timm

| Small-scale | | | | Large-scale | | | |
|---|---|---|---|---|---|---|---|
| Model | # param. | Type | Efficient | Model | # param. | Type | Efficient |
| dla46x_c | 1.1 | CNN | ✗ | res2next50 | 24.7 | CNN | ✗ |
| dla46_c | 1.3 | CNN | ✗ | resnext50d_32x4d | 25.0 | CNN | ✗ |
| mobilevitv2_050 | 1.4 | Other | ✓ | res2net50_14w_8s | 25.1 | CNN | ✗ |
| mobilenetv2_050 | 2.0 | CNN | ✓ | resnetv2_50 | 25.5 | CNN | ✗ |
| semnasnet_075 | 2.9 | Other | ✓ | resnetblur50 | 25.6 | CNN | ✗ |
| pvt_v2_b0 | 3.7 | Other | ✓ | resnetaa50 | 25.6 | CNN | ✗ |
| convnext_atto | 3.7 | CNN | ✗ | ecaresnet50t | 25.6 | Other | ✓ |
| mnasnet_100 | 4.4 | Other | ✓ | ecaresnet50d | 25.6 | Other | ✓ |
| spnasnet_100 | 4.4 | Other | ✓ | gcresnet50t | 25.9 | Other | ✗ |
| ghostnet_100 | 5.2 | CNN | ✓ | dla102x | 26.3 | CNN | ✗ |
| hardcorenas_a | 5.3 | Other | ✓ | xception41p | 26.9 | CNN | ✗ |
| efficientnet_b0 | 5.3 | CNN | ✓ | xception41 | 27.0 | CNN | ✗ |
| fbnetc_100 | 5.6 | CNN | ✓ | gluon_seresnext50_32x4d | 27.6 | CNN | ✗ |
| mobilevit_s | 5.6 | Other | ✓ | cspdarknet53 | 27.6 | Other | ✓ |
| tinynet_a | 6.2 | CNN | ✓ | legacy_seresnet50 | 28.1 | CNN | ✗ |
| ese_vovnet19b_dw | 6.5 | CNN | ✓ | repvgg_a2 | 28.2 | CNN | ✓ |
| densenet121 | 8.0 | CNN | ✗ | convnext_tiny_hnf | 28.6 | CNN | ✗ |
| densenetblur121d | 8.0 | CNN | ✗ | densenet161 | 28.7 | CNN | ✗ |
| crossvit_9_240 | 8.6 | Other | ✓ | ecaresnetlight | 30.2 | Other | ✗ |
| fbnetv3_b | 8.6 | CNN | ✓ | selecsls60 | 30.7 | CNN | ✗ |
| resnet14t | 10.1 | CNN | ✗ | gernet_l | 31.1 | CNN | ✓ |
| seresnext26ts | 10.4 | Other | ✗ | selecsls42b | 32.5 | CNN | ✗ |
| gcresnext26ts | 10.5 | Other | ✗ | selecsls60b | 32.8 | CNN | ✗ |
| eca_botnext26ts_256 | 10.6 | Other | ✗ | dla102 | 33.3 | CNN | ✗ |
| bat_resnext26ts | 10.7 | Other | ✗ | resnetrs50 | 35.7 | CNN | ✗ |
| lambda_resnet26rpt_256 | 11.0 | Other | ✗ | resnet51q | 35.7 | CNN | ✗ |
| resnet18d | 11.7 | CNN | ✗ | darknetaa53 | 36.0 | CNN | ✗ |
| halonet26t | 12.5 | Other | ✗ | resnet61q | 36.8 | CNN | ✗ |
| botnet26t_256 | 12.5 | Other | ✗ | dpn92 | 37.7 | CNN | ✗ |
| dpn68 | 12.6 | CNN | ✗ | xception65p | 39.8 | CNN | ✗ |
| dpn68b | 12.6 | CNN | ✗ | gluon_xception65 | 39.9 | CNN | ✗ |
| gc_efficientnetv2_rw_t | 13.7 | Other | ✓ | dla102x2 | 41.3 | CNN | ✗ |
| sehalonet33ts | 13.7 | Other | ✗ | xception71 | 42.3 | CNN | ✗ |
| sebotnet33ts_256 | 13.7 | Other | ✗ | twins_pcpvt_base | 43.8 | Other | ✗ |
| densenet169 | 14.1 | CNN | ✗ | gluon_resnext101_32x4d | 44.2 | CNN | ✗ |
| maxvit_nano_rw_256 | 15.5 | Other | ✗ | ecaresnet101d | 44.6 | Other | ✓ |
| gcresnet50ts | 15.7 | Other | ✗ | res2net101_26w_4s | 45.2 | CNN | ✗ |
| dla34 | 15.7 | CNN | ✗ | cs3edgenet_x | 47.8 | Other | ✓ |
| ecaresnet26t | 16.0 | Other | ✓ | gluon_seresnext101_32x4d | 49.0 | CNN | ✗ |
| resnet26d | 16.0 | CNN | ✗ | cs3se_edgenet_x | 50.7 | Other | ✓ |
| maxxvit_rmlp_nano_rw_256 | 16.8 | Other | ✗ | efficientnetv2_rw_m | 53.2 | CNN | ✓ |
| seresnext26t_32x4d | 16.8 | Other | ✗ | dla169 | 53.4 | CNN | ✗ |
| seresnext26d_32x4d | 16.8 | Other | ✗ | sequencer2d_l | 54.3 | Other | ✗ |
| dla60x | 17.4 | CNN | ✗ | poolformer_m36 | 56.2 | Other | ✗ |
| resnet32ts | 18.0 | CNN | ✗ | gluon_resnet152_v1b | 60.2 | CNN | ✗ |
| edgenext_base | 18.5 | Other | ✓ | resnet152d | 60.2 | CNN | ✗ |
| eca_resnet33ts | 19.7 | Other | ✓ | dpn98 | 61.6 | CNN | ✗ |
| seresnet33ts | 19.8 | Other | ✗ | resnetrs101 | 63.6 | CNN | ✗ |
| gcresnet33ts | 19.9 | Other | ✗ | resnet200d | 64.7 | CNN | ✗ |
| densenet201 | 20.0 | CNN | ✗ | seresnet152d | 66.8 | Other | ✗ |
| cspresnext50 | 20.6 | CNN | ✗ | wide_resnet50_2 | 68.9 | CNN | ✗ |
| regnetv_040 | 20.6 | CNN | ✗ | dm_nfnet_f0 | 71.5 | CNN | ✗ |
| convmixer_768_32 | 21.1 | CNN | ✗ | dpn131 | 79.3 | CNN | ✗ |
| cs3darknet_focus_l | 21.2 | CNN | ✗ | pnasnet5large | 86.1 | Other | ✗ |
| hrnet_w18 | 21.3 | CNN | ✗ | resnetrs152 | 86.6 | CNN | ✗ |
| cspresnet50 | 21.6 | Other | ✓ | dpn107 | 86.9 | CNN | ✗ |
| gluon_resnet34_v1b | 21.8 | CNN | ✗ | swinv2_base_window8_256 | 87.9 | Other | ✗ |
| resnet34d | 21.8 | CNN | ✗ | nasnetalarge | 88.8 | Other | ✗ |
| cs3sedarknet_l | 21.9 | Other | ✓ | resnetrs200 | 93.2 | CNN | ✗ |
| dla60 | 22.0 | CNN | ✗ | seresnext101d_32x8d | 93.6 | Other | ✗ |
| lamhalobotnet50ts_256 | 22.6 | Other | ✗ | seresnextaa101d_32x8d | 93.6 | Other | ✗ |
| halo2botnet50ts_256 | 22.6 | Other | ✗ | ecaresnet269d | 102.1 | Other | ✓ |
| halonet50ts | 22.7 | Other | ✗ | legacy_senet154 | 115.1 | CNN | ✗ |
| adv_inception_v3 | 23.8 | CNN | ✗ | resnetrs270 | 129.9 | CNN | ✗ |
| gluon_inception_v3 | 23.8 | CNN | ✗ | vgg11 | 132.9 | CNN | ✗ |
| | | | | vgg13 | 133.0 | CNN | ✗ |

