# OpenReview forum: "PePR: Performance Per Resource Unit as a Metric to Promote Small-scale Deep Learning in Medical Image Analysis"
_NLDL.org/2025/Conference — NLDL 2025 Oral_

### Official Review · Reviewer_5oEE · 2024-09-27

**Confidence:** 3

**Summary:**

The paper titled introduces the PePR score (Performance per Resource Unit) as a novel metric to evaluate deep learning (DL) models, particularly in resource-constrained settings like the Global South. The research focuses on balancing model performance with resource consumption (e.g., compute power, energy, and data). It argues that the trend of large-scale models is unsustainable, creates barriers for resource-limited researchers, and leads to inequity in AI development.

**Strengths:**

One of the key strengths of this paper is its introduction of the PePR score, a metric that effectively quantifies the trade-off between model performance and resource consumption, providing a holistic evaluation of deep learning models. This approach addresses a gap in AI research by promoting efficiency, especially for resource-constrained environments, which are often overlooked in conventional performance metrics. Additionally, the paper's comprehensive analysis of 131 different DL models across three medical imaging datasets offers valuable insights into the performance-resource dynamics, making the research both thorough and applicable to real-world scenarios. By highlighting the advantages of smaller, pretrained models, the study challenges the prevailing focus on large-scale models, pushing for a more sustainable and equitable approach to AI. This is particularly important for promoting AI accessibility in the Global South, enhancing the paper’s relevance and societal impact.

**Weaknesses:**

One weakness of the paper is that it primarily focuses on the evaluation of deep learning models in the context of medical imaging, which may limit the generalizability of the findings to other domains where the characteristics of datasets and model requirements differ significantly. Additionally, while the PePR score is a novel and useful metric, its effectiveness in optimizing models outside of small-scale or resource-constrained environments is not thoroughly explored, potentially reducing its applicability to high-performance use cases. The paper also relies heavily on pretrained models, and while it demonstrates their advantages, it does not fully investigate the potential trade-offs, such as the limitations in model flexibility or the dependence on the quality and diversity of the pretraining data. Moreover, the experiments are conducted over a relatively short training period (10 epochs), which might not fully capture the long-term performance trends and resource consumption patterns of the models. These factors could affect the broader applicability and robustness of the conclusions.

**Justification:**

The evaluation of the paper highlights both strengths and weaknesses, providing a balanced perspective on its contribution to deep learning research. The strengths lie in the introduction of the PePR score, which effectively promotes resource-efficient AI by balancing performance and resource costs—a particularly valuable contribution for resource-constrained regions. The paper’s comprehensive analysis of a wide range of models across multiple datasets, alongside its emphasis on small-scale models, pushes the AI community to reconsider the prevalent trend of large-scale, resource-heavy models. This makes the work not only innovative but also socially impactful, addressing issues of equity and sustainability in AI.

On the other hand, the weaknesses indicate that the paper’s scope is somewhat narrow, focusing heavily on medical imaging, which may limit the general applicability of its findings. The reliance on pretrained models and short training epochs raises concerns about the broader robustness of the PePR score, especially in domains with different data characteristics or where model flexibility and long-term resource consumption are critical.

Overall, I judge the paper as a valuable contribution to the field, especially for promoting sustainable deep learning practices. However, its impact could be further enhanced by extending its scope beyond medical imaging and offering a deeper exploration of the limitations of pretrained models and the PePR score in various real-world settings.

---

> ### Author Rebuttal · Authors · 2024-10-25
>
> We thank the reviewer for the feedback. We address their concerns below, and also with the revised manuscript.
>
> > One weakness of the paper is that it primarily focuses on the evaluation of deep learning models in the context of medical imaging, which may limit the generalizability of the findings to other domains where the characteristics of datasets and model requirements differ significantly.
> > Moreover, the experiments are conducted over a relatively short training period (10 epochs), which might not fully capture the long-term performance trends and resource consumption patterns of the models.
>
> We understand the concerns of the reviewer in this regard, and have adjusted the claims in the paper to be specific to medical imaging datasets and by clarifying this in the title. The title now reads "PePR: Performance Per Resource Unit as a Metric to Promote Small-scale Deep Learning for _**Medical Image Analysis**_".
>
> We have not experimented on other datasets outside of the medical image analysis domain, as the connection to equity was the most straightforward with this application domain. Future works could consider expanding this analysis to other domains, as the PePR score itself is agnostic to the downstream task.
>
> Furthemore, to show that our experimental results are not artifacts of the medical imaging dataset or of fine-tuning for 10 epochs, we consider the performance of models that have been trained to convergence on the ImageNet dataset (which formed the basis of pre-training) using the public dataset from Pytorch Image Models library. We perform a similar analysis of validation performance of the converged models and PePR-M score shown in (the updated) Figure 4-b), which are similar to the trends reported in our experiments in Figure 3. We observe that small scale models offer a better trade-off between performance and resource consumption, even when all the models are trained to convergence.
>
> We have clarified this in the Limitations sub-section in the revised paper, along with the updated Figure 4.
>
> > Additionally, while the PePR score is a novel and useful metric, its effectiveness in optimizing models outside of small-scale or resource-constrained environments is not thoroughly explored, potentially reducing its applicability to high-performance use cases.
>
> When high-performance is deemed to be more important than resource consumption, PePR score can be adjusted to reflect this. To achieve this we introduced the $\alpha$ parameterised version of PePR score resulting in $P_{ePR}(R,P;\alpha) = \alpha \cdot P/(\alpha + R)$ with $\alpha \ge 1$. Setting a large value, say with $\alpha=100$, would prioritise performance and disregard the effect of the resource consumption. As an example, consider the PePR score for the most resource intensive model that also achieves the best performance i.e., $P=1.0,R=1.0$. According to the definition in Eq. (1), the PePR score is $P_{ePR}(R=1,P=1;\alpha=1)=0.5$. However, by increasing the emphasis on performance using $\alpha=100$, now the $P_{ePR}(R=1,P=1;\alpha=100)=0.99$, basically ignoring the resource costs, if the application warrants this. Intermediate values of $\alpha$ can offer a spectrum of trade-off between how the resource costs should be taken into consideration. In this work, we are focused on operating in resource constrained regimes, and are mainly interested in the setting with $\alpha=1$.
>
> We have now included this discussion in the revised paper in Sec. 2.
>
>
> > The paper also relies heavily on pretrained models, and while it demonstrates their advantages, it does not fully investigate the potential trade-offs, such as the limitations in model flexibility or the dependence on the quality and diversity of the pretraining data.
>
> We agree that pre-training on a single dataset such as ImageNet might not be well-suited in all scenarios. In our experiments, use of existing pre-trained models showed improved performance when compared to training them from scratch.
>
> We will include this to be a limitation in Sec. 4.

---

### Official Review · Reviewer_uyFP · 2024-10-04
**Interesting problem, but more results would improve the clarity of the paper**

**Confidence:** 4

**Summary:**

This paper discusses the performance-resource tradeoff in small- and large-scale DL models. The authors shed light on the inaccessibility to large-scale computing and data, which hinders researchers' ability, especially in the global south and healthcare-AI sectors. The paper proposed a new metric that considers both performance and resources for DL models. An evaluation of more than 100 models for medical image classification datasets shows that small-scale models can perform similarly to large-scale models under resource-constrained regimes. The authors also claim that fine-tuning using a pretrained model should always be a first option rather than training from scratch.

**Strengths:**

- The paper is well-motivated from the angle of compute resource inaccessibility and how this can hinder the ability of some researchers to build high-performing models, especially for the sake of equity in AI-based healthcare systems.
- The proposed performance-resource tradeoff metric is interesting and can be incorporated into closed-loop optimization frameworks.
- The paper is well-written and easy to follow.

**Weaknesses:**

- Although the paper is well-motivated, the authors haven't discussed the results more profoundly or mapped their observations onto realistic cases. For instance, additional metrics on the usage and adoption of specific models/datasets for healthcare systems per region would be interesting to emphasize the resource equity barrier problem highlighted by the authors.
- The PePR metric's intended usage is still unclear. How do the authors envisage using this metric in a multi-objective optimization problem, and what would be the realistic use case?
- The process of models selection for the study is not well detailed. What criteria did the authors use to choose CNN and other models? Given the variety of model architectures, training methods, and task complexity, How could the authors ensure a representative coverage of models, tasks, and datasets so as to prevent any unbiased results? This needs to be justified further.
- The paper lacks additional details on the fine-tuning method and the rationale behind the training for 10-epoch. What if the fine-tuning is performed for less/more than 10 epochs? Would the paper observations/results still hold?
- In Figure 3, it's clear that CNN and others show no difference, so such categorization is not needed. The scatter points can reflect DL models in general.
- The discussion section could be more comprehensive (e.g., . what recommendation would the authors propose to tackle the equity problem)
- Ablation studies, particularly on fine-tuning methods and budget, models architectures and scale, and complexity of datasets, are crucial to confirm the generality of the authors' observations. This would further validate the robustness of their results.

**Final Rebuttal Confidence:**

4

**Final Rebuttal Justification:**

The authors provided detailed responses in the rebuttals, which addressed several of my concerns. The paper contains interesting results and insights worth sharing with the ML research community. Hence, I recommend accepting the revised version of the paper.

**Justification:**

The paper posits an interesting research problem regarding equity in resource access and how this would impact DL-based healthcare systems. However, the authors need to discuss the results from the lens of the problem and add more results to back up their claims. Further details on models selection and fine-tuning method/budget need to be considered to verify the result's robustness. The discussion section needs to emphasize the importance of the observations and discuss limitations and recommendations for the community.

---

> ### Author Rebuttal · Authors · 2024-10-25
>
> We thank the reviewer for their constructive feedback. We address their concerns with clarifications and revisions in the updated manuscript.
>
> > Although the paper is well-motivated, the authors haven't discussed the results more profoundly or mapped their observations onto realistic cases.
>
> Due to the initial page limit, we could not delve into many discussion points. In the revised version, we have updated the discussions to include additional discussion points, and some of the limitations of our work.
>
> These are now updated in Sec. 4.
>
> > .. additional metrics on the usage and adoption of specific models/datasets for healthcare systems per region would be interesting to emphasize the resource equity barrier problem highlighted by the authors.
>
> This is the first work where we have introduced PePR as a novel score that jointly considers performance and resource consumption. Using this score to analyse a broad family of models, we have concretely shown the differences in the better performance to resource consumption trade-off offered by small scale models.
>
> We do not have access to the usage of specific models/ datasets per region. We would be keen on knowing if the reviewer is aware of sources of this nature. As we believe collecting such information is an involved project in itself and could be considered in future work.
>
> > The PePR metric's intended usage is still unclear. How do the authors envisage using this metric in a multi-objective optimization problem, and what would be the realistic use case?
>
> PePR score is a composite metric that offers a trade-off between performance and resource consumption. It can be used instead of multi-objective optimisation of the two objectives separately.
>
> We envision the use of PePR score to compare models that use different extents of resources. Current reporting trends in deep learning for image analysis mainly focus on performance metrics like accuracy disregarding the resources expended. Furthermore, PePR can be used to choose the best model under a known resource constraint, such as maximum memory or energy consumption allowed.
>
> We include this in the updated Discussions section.
>
> > The process of models selection for the study is not well detailed. What criteria did the authors use to choose CNN and other models? Given the variety of model architectures, training methods, and task complexity, How could the authors ensure a representative coverage of models, tasks, and datasets so as to prevent any unbiased results? This needs to be justified further.
>
> We randomly sub-sampled the available models in Pytorch Image Models library, which during our experiments had about 700 models. We chose as many unique architectures as possible that were all pre-trained on the same ImageNet dataset. This resulted in the 131 models used in our work which spans CNN, vision transformers, hybrid models, and efficient architectures.
>
> We report the experiments on three diverse medical imaging datasets for image classification.
>
> To be more specific in our claims we adjust the title to specify "medical image analysis" and discuss these further in Sec. 3 on Data and in Sec. 4 under Limitations sub-section in the revised manuscript.
>
> >The paper lacks additional details on the fine-tuning method and the rationale behind the training for 10-epoch. What if the fine-tuning is performed for less/more than 10 epochs? Would the paper observations/results still hold?
>
> We consider training or fine-tuning of 10 epochs to reduce the compute resources used in this work. Training of all the 131 models considered to convergence for each of the datasets would be prohibitively expensive.
>
> However, we do agree this can be a limitation, as different models learn at different rates. To show that our experimental results are not artifacts of this choice, we consider the performance of models that have been trained to convergence on the ImageNet dataset (which formed the basis of pre-training) using the public dataset from Pytorch Image Models library. We perform a similar analysis of validation performance of the converged models and PePR-M score shown in (the updated) Figure 4-b), which are similar to the trends reported in our experiments in Figure 3. We observe that small scale models offer a better trade-off between performance and resource consumption, even when all the models are trained to convergence.
>
> We have updated these discussions in the revised paper, along with the new analysis in the Discussions section.
>
> > In Figure 3, it's clear that CNN and others show no difference, so such categorization is not needed. The scatter points can reflect DL models in general.
>
> We have removed these marker points in the plots in the revised manuscript.
>
> >The discussion section could be more comprehensive (e.g., . what recommendation would the authors propose to tackle the equity problem)
>
> We have expanded the Discussion section comprehensively now that includes discussion of several points raised by all reviewers.
>
> > Ablation studies, particularly on fine-tuning methods and budget, models architectures and scale, and complexity of datasets, are crucial to confirm the generality of the authors' observations. This would further validate the robustness of their results.
>
> The additional results with ImageNet, that has been trained to convergence hopefully alleviates some of these concerns.

---

### Official Review · Reviewer_HFyD · 2024-10-07
**I would suggest major revisions before acceptance.**

**Confidence:** 3

**Summary:**

The paper introduces the PePR score, a novel metric that measures performance per resource unit for deep learning (DL) models, with a focus on vision tasks, particularly in resource-constrained settings like medical imaging. The authors argue that large-scale DL models come with high costs in terms of compute, data, and energy, which can create barriers for researchers with limited access to these resources, especially in the Global South. The PePR score is defined in a balanced and objective way that accounts for both performance and resource costs. This method emphasizes resource sensitivity for high-performing models and highlights diminishing returns in resource usage.

**Strengths:**

The PePR score is defined in a balanced and objective way that accounts for both performance and resource costs. This method emphasizes resource sensitivity for high-performing models and highlights diminishing returns in resource usage.

**Weaknesses:**

Can you provide a more thorough justification for normalizing all resource types equally? Should different types of resources (e.g., memory vs. energy) be weighted differently depending on the application or hardware?

Have you considered evaluating PePR across different domains (e.g., NLP or general object detection tasks) to assess its broader applicability? Are there plans to extend the evaluation beyond medical image datasets?

Do you intend to extend PePR to consider other constraints like monetary cost, hardware access? How would the metric adapt in such contexts?

**Justification:**

While the PePR score is an innovative and important contribution to the discussion on resource efficiency in deep learning, the assumptions behind the PePR score, the narrow scope of datasets, and the absence of empirical comparisons with other resource-aware metrics reduce the immediate practical significance of the work.

---

> ### Author Rebuttal · Authors · 2024-10-25
>
> We thank the reviewer for their feedback. We address their concerns with clarifications and revisions in the updated manuscript.
>
> > Can you provide a more thorough justification for normalizing all resource types equally? Should different types of resources (e.g., memory vs. energy) be weighted differently depending on the application or hardware?
>
> In our formulation, we have presented PePR as a general framework which can consider any of the relevant resources.
>
> We believe that the reviewer is considering a scenario where the different resources can be jointly taken into account. We have not considered this setting, however, we can envision a general PePR score which can consider all resources into account by weighting them differently.
>
> $P_\text{ePR} = \frac{P}{1+\sum_i w_iR_i}$ with $\sum_i w_i = 1, R_i \in [0,1]$.
>
> As the reviewer suggests, the different weights can be adjusted depending on the application. In our work the key experiments are considering energy consumption as the main resource in the PePR-E score. Additional metrics (PePR-M for memory, PePR-C for carbon emissions, PePR-T for training time) reported in the Figure A.2 were to show the versatility of the PePR score.
>
> We have included this discussion in the revised paper.
>
> > Have you considered evaluating PePR across different domains (e.g., NLP or general object detection tasks) to assess its broader applicability? Are there plans to extend the evaluation beyond medical image datasets?
>
> We understand the concerns of the reviewer in this regard, and have adjusted the claims in the paper to be specific to medical imaging datasets and by clarifying this in the title. The title now reads "PePR: Performance Per Resource Unit as a Metric to Promote Small-scale Deep Learning for _**Medical Image Analysis**_".
>
> We have not experimented on other datasets outside of the medical image analysis domain, as the connection to equity was the most straightforward with this application domain. Future works could consider expanding this analysis to other domains, as the PePR score itself is agnostic to the downstream task.
>
> We have clarified this in the Limitations sub-section in the revised paper.
>
> > Do you intend to extend PePR to consider other constraints like monetary cost, hardware access? How would the metric adapt in such contexts?
>
> The simple formulation of PePR score allows for inclusion of additional resource factors. By normalizing to the maximum allowed resource (could be monetary costs), the relative use of these resources to improve performance will be captured by PePR score (as already shown in our analyses in Sec 2. on performance curves).

---

### Official Review · Reviewer_WeNX · 2024-10-09
**Review of PePR: Performance Per Resource Unit as a Metric to Promote Small-scale Deep Learning**

**Confidence:** 4

**Summary:**

In this work, the authors propose a metric called PePR, which is a measure of performance per resource-unit and use it to try to show that small-scale Deep Learning has a better trade-off of performance to resource cost. They do this by conducting an empirical experiment in which they train/fine-tune a large class of models and evaluate PePR on them.

**Strengths:**

* The PePR metric is highly relevant due to the concerns of resource costs of training machine learning models.
* Although the PePR metric is simple, it is highly interpretable (which the authors also walk through in the paper).

**Weaknesses:**

Major Weaknesses:
* There is one primary (large) reason for my low score which is that I am concerned about the fact that the authors only train all their models for 10 epochs across all experiments. I could be wrong, but it seems highly unlikely to me that models of the reported sizes are converged after 10 epochs of training _especially_ when trained from scratch. If they are not converged, then the conclusion to promote small-scale deep learning could be wrong. I would appreciate it if the authors would comment on this.

Minor Weaknesses:
* In the abstract the authors write that using pre-trained models may be more efficient, but (to me) this seems conflicting with the recommendation of using small-scale Deep learning. Pre-training large models is typically highly resource costly and I suppose not considered small-scale. Could the authors comment on this?
* Could the authors comment a bit on how to tune the proposed $\alpha$ in line 162?

Minor corrections/suggestions:
* I think there is a "the" missing in line 041 before medical.
* I think the definitive "the" in line 044 before energy should be removed for grammatical reasons.
* In line 073, it would be nice to write what "a novel measure" is a measure of.
* In Figure 2 (b) the x-axis goes from 0.000 to 0.015. Should it not be from 0.0 to 1.0 after normalization?
* There is a full stop missing in line 219 before "Other".
* Could the authors clarify what they mean about "efficient" versus "not efficient" in line 220-222?
* I think the paragraph in lines 341-349 should be move to the conclusion.

**Justification:**

I have provided a score of 2 primarily due to the concerns with convergence of the presented models which potentially could unravel/change the entire conclusion on model scale. My comments on tuning of $\alpha$ and especially on conflicting messages with regards to what model scale to use, are also a contributor to the score being a 2, but not to the same degree as the issues with model convergence.

---

> ### Author Rebuttal · Authors · 2024-10-25
>
> ### Major weakness:
>
> We thank the reviewer for their careful consideration of our work. We address their concerns with clarifications and revisions in the updated manuscript.
>
> > There is one primary (large) reason for my low score which is that I am concerned about the fact that the authors only train all their models for 10 epochs across all experiments. I could be wrong, but it seems highly unlikely to me that models of the reported sizes are converged after 10 epochs of training especially when trained from scratch. If they are not converged, then the conclusion to promote small-scale deep learning could be wrong. I would appreciate it if the authors would comment on this.
>
> We consider training or fine-tuning of 10 epochs to reduce the compute resources used in this work. Training of all the 131 models considered to convergence for each of the datasets would be prohibitively expensive.
>
> However, we do agree this can be a limitation, as different models learn at different rates. To show that our experimental results are not artifacts of this choice, we consider the performance of models that have been trained to convergence on the ImageNet dataset (which formed the basis of pre-training) using the public dataset from Pytorch Image Models library. We perform a similar analysis of validation performance of the converged models and PePR-M score shown in (the updated) Figure 4-b), which are similar to the trends reported in our experiments in Figure 3. We observe that small scale models offer a better trade-off between performance and resource consumption, even when all the models are trained to convergence.
>
> We have updated these discussions in the revised paper, along with the new analysis in the Discussions section and updated Figure 4.
>
> ### Minor weaknesses:
>
> > In the abstract the authors write that using pre-trained models may be more efficient, but (to me) this seems conflicting with the recommendation of using small-scale Deep learning. Pre-training large models is typically highly resource costly and I suppose not considered small-scale. Could the authors comment on this?
>
> We would like to clarify here that by pre-trained models, we specifically are talking about _existing_ pre-trained models that are publicly available. According to our experiments, using _already_ pretrained models on ImageNet is better than training from scratch. If pre-trained models do not exist, then we do not recommend training on additional datasets to reduce resource consumption.
>
> We clarify this in the Abstract and rest of the paper in the revised version.
>
> > Could the authors comment a bit on how to tune the $\alpha$ proposed in line 162?
>
> When performance is deemed to be more important than resource consumption, PePR score can be adjusted to reflect this. To achieve this we introduced the $\alpha$ parameterised version of PePR score resulting in $P_{ePR}(R,P;\alpha) = \alpha \cdot P/(\alpha + R)$ with $\alpha \ge 1$. Setting a large value, say with $\alpha=100$, would prioritise performance and disregard the effect of the resource consumption. As an example, consider the PePR score for the most resource intensive model that also achieves the best performance i.e., $P=1.0,R=1.0$. According to the definition in Eq. (1), the PePR score is $P_{ePR}(R=1,P=1;\alpha=1)=0.5$. However, by increasing the emphasis on performance using $\alpha=100$, now the $P_{ePR}(R=1,P=1;\alpha=100)=0.99$, basically ignoring the resource costs, if the application warrants this. Intermediate values of $\alpha$ can offer a spectrum of trade-off between how the resource costs should be taken into consideration. In this work, we are focused on operating in resource constrained regimes, and are mainly interested in the setting with $\alpha=1$.
>
> We have now included this discussion in the revised paper (Sec. 2).
>
> ### Minor corrections/ suggestions:
>
> We thank the reviewer for their meticulous suggestions. We have corrected all the relevant instances. The normalised energy in Fig. 2 (b) is 0.015 as the maximum of 1.0 is due to another, much larger model, resulting in ESE-VoVNet with a maximum energy of 0.015.

---

### Meta-Review · Area_Chair_rJAR · 2024-11-01

**Recommendation:** Accept (Oral)
**Confidence:** 5

**Metareview:**

This paper introduces the PePR score, a metric designed to evaluate the trade-off between performance and resource consumption in deep learning models, with a particular focus on resource-constrained environments like medical imaging. The study is timely, tackling AI equity concerns by highlighting the resource efficiency of smaller models. The PePR score offers a novel, interpretable way to quantify resource use across various DL models, making a positive impact in promoting sustainable practices in AI.

However, the study’s reliance on short training durations and a specific domain (medical imaging) limits its broader applicability. Furthermore, the absence of statistical validation, alongside the reliance on pre-trained models, raises concerns about the robustness and flexibility of the findings. While the PePR score is a promising metric, extending evaluations to varied datasets, domains, and training regimes would strengthen its credibility and adoption potential.

Overall, the paper presents a valuable contribution, offering insights into efficient DL practices in resource-constrained settings, but would benefit from further validation and refinement to maximize its impact.

**Suggested Changes To The Recommendation:**

1: I agree that the recommendation could be moved down

---

### Decision · Program_Chairs · 2024-11-06

**Decision:**

Accept (Oral)

**Comment:**

We recommend an oral and a poster presentation given the AC and reviewers recommendations.